# Epigenetics in Families: Covariance between Mother and Child Methylation Patterns

**DOI:** 10.3390/brainsci11020190

**Published:** 2021-02-04

**Authors:** Tanya Van Aswegen, Guy Bosmans, Luc Goossens, Karla Van Leeuwen, Stephan Claes, Wim Van Den Noortgate, Benjamin L. Hankin

**Affiliations:** 1Department of Psychiatry, University of Stellenbosch, 7505 Cape Town, Tygerberg, South Africa; t.vanaswegen@vu.nl; 2Department of Clinical Psychology, VU University Amsterdam, 1081 BT Amsterdam, The Netherlands; 3Clinical Psychology, Faculty of Psychology and Educational Sciences, KU Leuven, 3000 Leuven, Belgium; 4School Psychology and Development in Context, Faculty of Psychology and Educational Sciences, KU Leuven, 3000 Leuven, Belgium; luc.goossens@kuleuven.be; 5Parenting and Special Education, Faculty of Psychology and Educational Sciences, KU Leuven, 3000 Leuven, Belgium; karla.vanleeuwen@kuleuven.be; 6University Psychiatric Center, Department of Neurosciences, KU Leuven, 3000 Leuven, Belgium; stephan.claes@kuleuven.be; 7Faculty of Psychology and Educational Sciences, KU Leuven Campus Kulak Kortrijk, 8500 Kortrijk, Belgium; wim.vandennoortgate@kuleuven.be; 8ITEC, IMEC Research Group at KU Leuven, 8500 Kortrijk, Belgium; 9Department of Psychology, University of Illinois at Urbana-Champaign, Champaign, IL 61820, USA; hankinb@illinois.edu

**Keywords:** DNA methylation, epigenetics, stress-related genes, shared environments, early adolescence

## Abstract

Theory and research both point at epigenetic processes affecting both parenting behavior and child functioning. However, little is known about the convergence of mother and child’s epigenetic patterns in families. Therefore, the current study investigated epigenetic covariance in mother–child dyads’ methylation levels regarding four stress-regulation related genes (*5HTT*, *NR3C1*, *FKBP5,* and *BDNF*). Covariance was tested in a general population sample, consisting of early adolescents (*M_age_* = 11.63, *SD_age_* = 2.3) and mothers (*N* = 160 dyads). Results showed that mother and offspring *5HTT* and *NR3C1* methylation patterns correlated. Furthermore, when averaged across genes, methylation levels strongly correlated. These findings partially supported that child and parent methylation levels covary. It might be important to consider this covariance to understand maladaptive parent–child relationships.

## 1. Introduction

In recent years, researchers have become increasingly aware of epigenetic changes, that is, changes in the expression of genes due to environmental influences (i.e., pesticides, bacteria, basic nutrients, etc.), which have a strong impact on people’s health, behavior, and disease [1]. Although epigenetic effects have been found in both parent and child development [2], little is known about potential epigenetic covariance at the level of mother–child dyads. Several mechanisms are known to epigenetically regulate genetic expression. One popular epigenetic mechanism is DNA methylation; methyl groups are added to the promoter region of genes, changing gene expression by blocking transcription factors’ access to the promoter region of the gene [3]. Methylation levels are affected by environmental stress, toxins [4], diet [5], and they influence—amongst others—the stress-regulation system [6]. The difficulty of regulating stress is known to increase maladaptive parenting [7] and increase child behavioral problems [8]. Maladaptive parenting and child behavior problems are known to be robustly correlated [9]. Since epigenetic effects have been reported in both adults’ and children’s problematic behavior, this raises the question of whether parents and children could have similar levels of methylation of stress-regulation related genes. However, research on DNA methylation covariance in parent child dyads is largely lacking.

Nevertheless, current theories of epigenetics imply covariance between parent and child methylation [10]. First, both mothers and children are exposed to the same environmental factors known to affect epigenetic changes peri- and post-partum [11]. They have a very similar diet and are to some extent exposed to the same stressors, such as the same bacteria and pollution. In addition, variation in methylation levels differs depending on genotypes that parents and children share [12]. This enhances the likelihood that their methylation levels would covary. Second, both theoretical and animal models suggest that epigenetic patterns might be inherited and passed on from parent to offspring [13]. Research shows that the methylation in some genes can be duplicated during mitosis [14]. Although the underlying mechanisms are still the focus of research, some animal studies do suggest that offspring might inherit epigenetic patterns that parents developed in response to certain environments [15]. In sum, it seems reasonable to assume that parents and children might have comparable methylation levels. Finding evidence for parent–child methylation covariance might contribute to our understanding of the complex processes explaining parent–child interactions. To our knowledge, just a single study investigated the correlation between parent and child methylation levels. This study found a small but significant correlation in Holocaust survivors [16]. However, the study had four key limitations: (a) it focused on methylation in a single gene (FK506 binding protein or *FKBP5*), (b) it relied on a small sample (*N* = 22 dyads only), (c) offspring methylation was measured in adulthood, and (d) surviving the holocaust is a very specific and severe type of distress that is difficult to generalize to other populations.

The aim of the current study was to remedy some of the shortcomings of the previous study of Yehuda et al. [16]—specifically, examining methylation covariance between mothers and children in several genes, in a larger sample, with offspring methylation measured at a younger age, and with participants from the general population presumably exposed to normative levels of daily distress. The covariance of methylation patterns for four genes related to the stress-regulation system was tested. The stress regulating system is regulated by the Hypothalamus–Pituitary–Adrenal (HPA) axis. Stress activates the HPA axis, which releases glucocorticoids (such as cortisol) to protect the body from harm by promoting fight–flight responses to distress. The HPA axis gets deactivated when the glucocorticoids activate glucocorticoid receptors (GR) in the hippocampus, which decreases arousal as one element of adequate stress regulation [17]. Methylation of the genes that manage this stress-regulation system can modulate levels of cortisol in response to stress, which translates into variation in self-regulation [18]. In the current study, we tested the correlation between mother’s and children’s methylation of these genes. More specifically, we looked at the methylation of 5HTT, *NR3C1*, *FKBP5*, and *BDNF* [19]. The serotonin transporter (*5HTT)* re-introduces the neurotransmitter serotonin from the synaptic cleft into presynaptic neurons [20]. The glucocorticoid receptor (*NR3C1)* is among the most studied GR genes and functions as a transcription factor that binds to glucocorticoid response elements in the promoters of glucocorticoid-responsive genes to inhibit HPA axis activity [21]. *FKBP5* acts as a co-chaperone that modulates, amongst others, glucocorticoid receptor activity in response to stressors [22]. Finally, the brain-derived neurotrophic factor (*BDNF*) regulates the development of the nervous system and the formation of appropriate synaptic connections in particular. BDNF expression occurs in the hippocampus, hypothalamus, and pituitary [23,24,25], which are three structures highly involved in the HPA axis activity and regulation. For all the above-mentioned genes, theory suggests that more exposure to adverse environments leads to higher levels of methylation, which may modulate the regulation negative affect and self-regulation [26].

To test the hypothesis that mother and child methylation levels for these genes are correlated, we used data of the ** study (Genes, Environment, Mood [27]), which focused on biological and family processes in the development of children’s depression. First, we investigated the covariance between mother and child methylation at the level of individual genes. Second, we looked at covariance between their overall methylation levels across these genes. So far, no research has investigated whether different environmental influences have specific effects on selected genes. Therefore, we tested the correlation between the factor scores that summarized the methylation scores across the four genes for mothers and children separately. Finally, although children in the GEM study were on average 11 years old, they ranged in age from 8 to 16 years. Since children increasingly spend more time in other environments than parents do during adolescence, older children might be exposed to different stressors or toxins than parents are. As a result, one could expect the covariance between mothers and children to decrease as children grow older. Therefore, we tested whether age moderated the covariance between mother and child methylation. Finally, we investigated the extent to which recent stress accounted for the levels of maternal and child methylation and affected their covariance.

## 2. Method

### 2.1. Participants

The sample comprised 160 dyads (mother and child; 42.5% boys and 57.5% girls). Children ranged in age from 8 to 16 years (*M* = 11.63, *SD* = 2.3), and parents’ age was unknown. This was a subsample of the larger GEM study sample for which both mother and child methylation data were available. Recruitment took place at two sites: the University ** and ** University. Parent reported that both the mother and child were fluent in English. Moreover, participating children did not have an autism spectrum or a psychotic disorder. All children had an IQ above 70. The sample was comparable to the community and school districts from which recruitment took place. The sample sufficiently represented the ethnic and racial characteristics of the overall population of the United States: 81.6% non-Hispanic; 18.4% Hispanic; 2% American Indian; 2% Asian; 5% Black, 75% White; 9% other; 7% multi-racial.

### 2.2. Measures

#### 2.2.1. Methylation Procedure 

To determine DNA methylation levels, we used quantitative bisulfite pyrosequencing by EpigenDx. Pyrosequencing for allele quantification (PSQ H96A, Qiagen Pyrosequencing) is a real-time sequencing-based DNA analysis. This analysis quantifies multiple and consecutive CpG sites (regions of DNA where a cytosine nucleotide is followed by a guanine nucleotide) individually as artificial T/C SNPs (a single nucleotide polymorphism) [28,29]. In short, 500 ng of sample DNA was bisulfate treated with the Zymo DNA Methylation Kit (Zymo Research, Orange, CA, USA). Bisulfate treated DNA is eluted in 20 μL volume, and 1 μL of it is used for each PCR. For the PCRs, we used the standard pyrosequencing recommended PCR condition: 10× PCR buffer, 1.5 mM MgCl_2_, 200 μM of each dNTP, 0.2 μM each of forward and reverse primers, HotStar DNA polymerase (Qiagen Inc.) 1.25 U, and 1 μL of bisulfite converted DNA per 30 μL reaction. PCR cycling conditions were 94 °C 15 min; 45 cycles of 94 °C 30 s; 46 °C 30 s; 72 °C 30 s; 72 °C 5 min; then, products were held at 4 °C. We used the Pyrosequencing PSQ96 HS System (PSQ H96A, Qiagen Pyrosequencing, Colorado, USA) to sequence the PCR products (each 10 μL). During this process, we fully followed the manufacturer’s instructions. Each locus’ methylation status was analyzed individually as a T/C SNP using QCpG software (PSQ H96A, Qiagen Pyrosequencing, Colorado, USA) [30].

For *NR3C1* methylation, human GCR Exon1F methylation assays covers 39 CG dinucleotides in the promoter region ranging from approximately −630 to −354 from the transcriptional start site based on Ensembl ID ENST00000231509. ADS749FS covers 7 CpG sites in the Exon1F region. For *5HTTP* methylation, human solute carrier family 6 (serotonin transporter) methylation assays covers 20 CG dinucleotides sites in the promoter reverse strand region ranging from approximately −69 to −213 from the transcriptional start site based on Ensembl ID ENSG00000108576. ADS580FS2 covers 16 CpG sites in the promoter region. For *FKBP5* methylation, human FK506 binding protein 5 methylation assays covers 10 CG dinucleotides sites in the promoter reverse strand region ranging from approximately −3092 to −3020 from the transcriptional start site based on Ensembl ID ENSG00000096060. ADS2500FS2 covers all 10 CpG sites in the promoter region. For *BDNF* methylation, human brain-derived neurotrophic factor (BDNF) methylation assays covers 8 CG dinucleotides sites in the Intron 1 region ranging from approximately +61111 to +61129 from the transcriptional start site based on Ensembl ID ENSG00000525950. ADS2107-FS covers all 8 CpG sites in the Intron 1 region.

#### 2.2.2. Recent Stress

To measure stress in the child, we administered two measures. First, we administered the Youth Life Stress Interview to the children (YLSI [31]). The YLSI is a reliable, semi-structured contextual stress interview that validly assesses ongoing stress level in adolescents. The YLSI’s reliability and validity is excellent [31,32]. Coding occurred in line with the procedure described by Rudolph and Flynn [31]. This is the procedure that is most often used. To derive stress severity information, responses to the related questions were coded by a team of three or more blind raters. They came to an agreed upon severity score. This score could range from 1 (little/no stress), 2 (average/normal stress), 3 (moderate stress), 4 (serious stress), to 5 (severe stress). The interview results in sum scores for stress severity and chronicity on several domains related to for example financial or academic stress. Second, we administered the Adolescent Life Events Questionnaire (ALEQ [33]). This questionnaire is designed to measure the occurrence of a broad range of negative events that typically occur during adolescence. For example, this included school problems (e.g., “You got in trouble with the teacher or principal”) or relationship difficulties (e.g., “You found out your boyfriend/girlfriend was cheating on you”). Adolescents rated each of the 37 events for frequency in the past 3 months on a Likert scale ranging from A (never) to E (always). Reliability and validity for the ALEQ has been established in past studies [34,35,36]. The three indicators of stress in the child were highly correlated (0.46 < *r* < 0.60; *ps* < 0.001), suggesting that it was better to combine all measures to reduce the number of analyses. To combine both stress measures, we calculated a sum score to express general child stress with higher scores indicating more stress. To measure stress in the parent, we administered an adult version of the ALEQ to calculate one sum score expressing parent stress.

#### 2.2.3. Procedure 

Saliva samples were collected from all study participants using the Oragene (DNA Genotek, Ottawa, ON, Canada) collection kits. The stress measures were administered at the time when the saliva was collected. All study measures and procedures were approved by University of Denver’s Internal Review Board.

## 3. Plan of the Analysis

All analyses were conducted using SPSS, version 27. First, we conducted preliminary analyses to evaluate whether child gender and age were directly linked to child methylation levels. Then, for Research Question 1, we calculated correlations to assess mother–child methylation covariance per gene. For Research Question 2, we used principal component analysis (PCA) on the four gene’s methylation scores for children and mothers separately. These PCAs allowed calculating a methylation component score per member of the dyad. Then, we assessed with a bivariate correlation analysis whether there is a general association between mother and child methylation independent of the specific genes. For Research Question 3, we evaluated the moderating effect of age on the association between mother and child methylation levels using Hayes’s [37] PROCESS tool for SPSS (Model 1). Separate analyses were performed for each individual gene and for the methylation component score. For Research Question 4, we first conducted correlation analyses to evaluate the association between maternal stress and maternal methylation levels, and then between child stress and child methylation levels. Finally, we conducted multiple regression analyses to evaluate the association between maternal and child methylation levels while controlling for the mother and child stress scores.

## 4. Results


*Preliminary Analyses: Gender and Age Effects*


No methylation data were missing. For 30 children, interview data were partially incomplete. When calculating the sum score, we used mean imputation. After that, less than 3% of the data was missing (one child stress score, four parent stress scores), which were pairwise deleted. Effects of children’s age and gender were tested separately. No significant correlations were found between children’s age and methylation. Analysis of variance indicated that also child gender was not significantly linked to methylation.

Research Question 1: Covariance at the Level of Individual Genes.

Table A1 of Appendix A shows significant correlations between the methylation levels of *5HTT, NR3C1*, *FKBP5*, and *BDNF* within each member of the dyad. In addition, the results showed significant correlation between mother and child methylation levels. However, Bonferroni correction for multiple testing requires *p*-values to be lower than 0.0125 (α/4 tests), due to which the transgenerational correlations for *FKBP5* and *BDNF* no longer reached significance (*p* = 0.044 and *p* = 0.046 respectively).

Research Question 2: Covariance of Methylation across Genes

Principal component analyses (PCAs) were conducted for children and mothers separately on the methylation scores for the four genes. PCA for the children resulted in one component with an eigenvalue higher than 1 (2.25), explaining 56% of the variance, and revealing factor loadings >0.60. PCA for the mothers resulted in one component with an eigenvalue higher than 1 (2.90), explaining 74% of the variance, revealing factor loadings >0.83. The adequacy of combining methylation levels across genes was further supported by reliability analysis (Cronbach α children = 0.72; Cronbach α mothers: 0.87).

Using the results of the PCAs, we computed factor scores for children and mothers separately. We found a significant correlation between the two factor scores, *r*(160) = 0.37, *p* < 0.001, reflecting a moderate association between child and mother methylation levels.

Research Question 3: Moderating Effect of Children’s Age on Mother–Child Methylation Covariance

To investigate moderation by age in the relationship between maternal and children’s methylation patterns, we first looked at the interaction for each of the four individual gene and then moved on to investigate the interaction for the methylation across genes component. No moderating effects for age were found for *NR3C1* (*b* = 0.07, *t*(156) = 1.28, *p* = 0.20); *5HTTP* (*b* = −0.03, *t*(156) = −0.63, *p* = 0.53); *FKBP5* (*b* = −0.05, *t*(156) = −1.02, *p* = 0.31) and *BDNF* (*b* = −0.01, *t*(156) = −0.11, *p* = 0.91). In addition, age did not moderate the relationship between mother and child (*b* = −0.07, *t*(156) = −1.27, *p* = 0.21).

Research Question 4: Stress and Mother–Child Methylation Covariance

Preliminary inspection of the stress scores showed that they had a skewed distribution. For mothers’ stress: *M* = 4.58; *SD* = 3.42; Skewness = 86, *SE*(Skewness) = 0.19; Kurtosis = 0.44, SE(Kurtosis) = 0.38. For children’s stress: *M* = 157.79; *SD* = 22.48. Moreover, data distribution was skewed: Skewness = −1.13, *SE*(Skewness) = 0.19; Kurtosis = 5.10, *SE*(Kurtosis) = 0.38. First, we calculated correlations between each mother’s and children’s recent stress and their methylation levels both at the level of each individual gene and at the component level. Table A2 shows only one significant correlation: stress of the child was related to higher child BDNF methylation levels. However, Bonferroni correction for multiple testing requires *p*-values to be lower than 0.01 (α/5 tests), due to which the correlation between *BDNF* methylation and child stress no longer reached significance (*p* = 0.025). Moreover, this correlation was no longer significant after excluding scores differing more than two *SD*s from the child stress mean, *r* = 0.05, *p* = 0.51.

Second, we used multiple regression analysis to test whether mother–child methylation covariance survived controlling for mother and child stress. The mother–child methylation associations remained significant for all the analyses: *5HTT*: *R*^2^ = 0.06, *F*(3, 152) = 2.98, *p* = 0.03, with *β_5HTTmother_* = 0.20, *p* = 0.04; *NR3C1*: *R*^2^ = 0.13, *F*(3, 152) = 7.62, *p* < 0.001, with *β_NR3C1mother_* = 0.33, *p* < 0.001; *FKBP5*: *R*^2^ = 0.03, *F*(3, 152) = 4.10, *p* = 0.008, with *β_FKBP5mother_* = 0.16, *p* = 0.044; *BDNF*: *R*^2^ = 0.08, *F*(3, 152) = 7.62, *p* < 0.001, with *β_BDNFmother_* = 0.15, *p* = 0.43; methylation component scores: *R*^2^ = 0.16, *F*(3, 152) = 9.94, *p* < 0.001, with *β_methylation_mother_* = 0.36, *p* < 0.001.

## 5. Discussion

This study investigated covariance between children’s and mothers’ levels of methylation for stress-regulation system-related genes. At the level of single genes, methylation in all four genes significantly correlated across the mothers and the children. However, the correlations for *FKBP5* and *BDNF* did not survive Bonferroni correction. The results of the current study regarding *FKBP5* were weaker than what Yehuda et al. [16] observed in Holocaust survivors. One reason why the size of the effect found in the current study was smaller could be the fact that the current study’s dyads were not exposed to the extreme distress that Holocaust survivors experienced. Nevertheless, the trend-like effect we found in the current study is in line with Yehuda’s study showing transgenerational covariance between mother and child methylation levels. Moreover, the current study added to the literature by investigating mother–child methylation covariance for additional genes and by investigating covariance at the same point in time. In line with expectations, methylation levels correlated between generations (mother–child) for *HTT5* and *NR3C1*. Results also showed that within the children and within the mothers, methylation between the genes correlated. In addition, results showed a significant correlation between mothers’ and children’s methylation component scores. Against our expectations, we found no evidence suggesting that the covariance between mothers and children declined as children grow older.

With regard to the covariance between mother and child methylation levels, correlations robustly supported our hypothesis. These findings suggest that the environment and the genetic background shared by children and mothers similarly affect the methylation patterns of both members of the dyad. The current findings call for more research to explain the causes of this covariance. Most straightforwardly, the covariance might reflect the immediate impact of the environment at the time of the study [38]. However, there was hardly any support for a correlation between methylation levels and stress. One reason for the lack of correlations between the stress measures and the methylation measures might be lack of variance at the level of stress in the current sample. Supporting this interpretation, the one observed association between child stress and BDNF methylation disappeared when removing more extreme scores. Moreover, ours is not the first study that struggled to find clear correlations between stress and methylation levels in humans [39]. In contrast to what is known about the epigenetic impact of chemical exposures, more research is needed to understand how and when stress exposure affects methylation [40]. For example, it might be that the shared methylation within each mother–child dyad we found in the current study reflects the ongoing effect of shared past exposure to chemical exposures [41]. Furthermore, recent research on the epigenetic clock [42] and on the Developmental Origins on Health and Disease model [43] suggest that methylation levels are to a substantial extent established earlier in life, making it harder to find significant correlations between ongoing stress and methylation levels at later ages. In addition, we could not correct for genetics in the analyses. Genotype can affect the likelihood that genes are methylated. Therefore, it might be that we could not find associations between stress and methylation because methylation levels and methylation covariance were driven by genetic covariance [12], which is an explanation we cannot rule out in the current study because we had no genetic data. Harder to test but impossible to rule out is the possibility that the covariance could also reflect epigenetic inheritance [44].

The fact that methylation levels correlated significantly within mothers and children and the fact that we also found evidence for covariance between mother and child methylation levels across all genes suggests that the factors affecting methylation might not be gene-specific. Instead, one possibility might be that shared exposure to a toxic versus healthy environment does result in similar methylation levels for different genes in both the mothers and the children. So far, little research investigated whether specific toxins affect methylation levels of specific genes. More research is needed to test whether methylation patterns are gene-specific or not.

For our hypothesis that age would moderate the association between mother and child methylation levels, we found no support. This suggests that the covariance between mothers’ and children’s methylation levels is not altered when their environments increasingly start to differ. It might be that the age range in the current study was too limited and that such age effects could emerge if children enter into adulthood themselves. Nevertheless, these results suggest that the covariance we identified in the current study was established earlier in life. One critical period could be during pregnancy or immediately post-partum, which is a time when children are considered most vulnerable to environmental influences [45]. Existing findings highlight the critical impact of prenatal environmental factors on DNA methylation as a mechanism by which early childhood experiences may become biologically embedded [46,47,48]. Another explanation might be that the covariance expresses at least to some extent epigenetic inheritance. However, recent attempts to establish epigenetic inheritance in humans were little successful [49,50]. Therefore, it is too premature to firmly draw such conclusions based on our data.

Further motivating caution when interpreting the findings is the small but significant association we found between recent child stress and the children’s BDNF methylation. This suggests that recent stress continues to have an effect on methylation levels, which could eventually lead to reduced mother–child methylation covariance later in life. Nevertheless, it is important to note that the child–mother covariance for BDNF methylation remained significant after accounting for the current stress effect and to note that the significant correlation did not survive Bonferroni correction. Therefore, this could just reflect statistical coincidence. In addition, to better evaluate whether mother and child methylation levels diverge when children grow older, it might be better for future research to account for more adverse sources of recent stress. For example, Romens et al. [51] found that child NR3C1 methylation levels were linked with exposure to physical maltreatment, displaying a specific change in the NR3C1 receptor gene. Hence, accounting for recent physical maltreatment could have decreased the mother–child covariance in the current study. In sum, more research is needed to better understand the nature of the mother–child covariance we demonstrated in this study.

Although the current findings provide some first evidence that studying methylation patterns in human dyads could move epigenetic research significantly forward, the interpretation of the findings should consider important limitations. First, families consist of more than two family members. Therefore, a more encompassing study of family epigenetics should take into account father and sibling information as well. Research has shown differential effects of maternal versus paternal contribution to offspring methylation [52]. Secondly, in the current study, there was a substantial lack of information about the current rearing environment. It would have been important to know more about the number of siblings and the composition of the family. Higher number of siblings and families with divorced parents, single parents, or recomposed families are known to be more stressful for children, and this could have better explained the mother–child covariance. In addition, there was a lack of information on the mothers’ environment and medical conditions before and during pregnancy, and on children’s current medical conditions such as hypercortisolism or other endocrinological abnormalities that could have affected the covariance. In addition, it would have been important to know maternal age as well as the mothers and children’s broader mental health conditions. Accounting for this information would have allowed giving a more precise indication of the mother–child epigenetic covariance in this sample. However, at first sight, it seems that these sources of variation should have suppressed the effect rather than artificially increased the effect. So, it seems promising that we still found such a substantial correlation. Since we cannot rule out that covariance might have been inflated by such factors, we think it is safest to say that the current study represents a new important step in the study of epigenetic patterns within families and that the current findings should motivate the future setup of better designed studies aimed at further unraveling mother–child covariance. Third, we could not present valid information on the mothers’ and children’s self-regulatory capacity nor on the participants’ cortisol responses to stress. Such information would have further strengthened the point that transgenerational epigenetic covariance might explain part of the transmission stress-regulation difficulties in parent–child dyads. Mothers did report on their children’s inhibitory control using the Early Adolescent Temperament questionnaire [53], but unfortunately, the internal consistency was insufficient as reported previously [53]. Fourth, we only focused on the stress-response system, and our findings might not generalize to epigenetics as applied to other biological systems. Finally, data collection was cross-sectional in nature. As a result, it is impossible to draw conclusions about causal mechanisms nor about any synergistic interactions between children and mothers’ methylation patterns.

In sum, the current study found evidence for covariance in mothers’ and children’s methylation levels of stress-regulation related genes. This finding could prove relevant, because methylation covariance could be one understudied factor explaining the previously established covariance between mothers’ and children’s ability to regulate distress [54,55]. For example, a better understanding of the role of epigenetics in the association between problematic mother and child self-regulatory behavior could improve our understanding of treatment-resistant transgenerational vulnerabilities found in multi-problem families [56] and the therapeutic needs of family members that developed unhealthy family relationships [57].

## 6. Conclusions

This study investigated the covariance between children’s and mothers’ levels of methylation for stress-regulation system-related genes. Results supported the transgenerational covariance for 5HTT and NR3C1 and for methylation component scores derived from all the methylation data across the genes we tested. More research is needed to evaluate the robustness of these findings and to uncover why parent–child methylation levels covary. Future research should investigate whether this covariance can help understand parent–child dynamics and child development. For example, parent–child methylation covariance could contribute to explain well-known transgenerational correlational patterns within families such as the association between parenting and the development of children’s emotional and behavioral problems. It might be that the shared methylation of genes involved in stress regulation might increase inadequate parenting (that requires well-developed stress-regulation capacities) and at the same time increase children’s emotional and behavior problems (which are also a marker of stress-regulation problems). Similarly, parent–child methylation covariance could help explain phenomena such as biobehavioral synchrony, resilience transmission, and transgenerational transmission in other domains of development (e.g., attachment). Clinically, more research similar to this could help soften professional caregivers’ blame-the-parent-like views on parenting and child psychopathology, because is it hard to blame parents for methylation covariance. A more positive view on parents who struggle with a child displaying stress-regulation problems can eventually improve therapist alliance and treatment outcomes.

## Data Availability

All data can be obtained from the corresponding author.

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
