# Peer review of "Epigenetics in Families: Covariance between Mother and Child Methylation Patterns"

_brainsci, 2021, doi:10.3390/brainsci11020190_

Round 1

Reviewer 1 Report

The authors report on cross-sectional associations between methylation of four stress-regulation related genes (5HTT, 14 NR3C1, FKBP5, and BDNF) in mothers and their children (9-16 yrs). In addition to basic correlations, the authors report on differential associations by child age and parent- and child perceived stress. Findings will be useful tor researchers studying epigenetics of parenting and child stress regulation. As such, the paper offers a modest but nonetheless interesting expansion on existing literature.

Throughout the manuscript, statements need to be checked on accuracy, precision, and nuance. A few examples:

- “to remedy all of the shortcomings” sounds perhaps a little to absolute. This study too, focuses on a limited set of genes, and its sample is relatively small. Perhaps “to remedy some of the shortcomings” would be more accurate.

- shared environment is not necessarily the most obvious explanation for concurrence. Biological dyads are likely to share genetics and this can drive methylation similarities given that methylation is also associated with genotype. More equal weighting could be given to the various explanations presented.

- The authors state they found “limited evidence that methylation were correlated with stress levels” In my view the evidence was less than limited, there was hardly any.

- In prior literature, higher methylation is not associated with poorer stress regulation and/or adversity exposure at all four genes (specifically FKBP5). It thus is not correct to state that in general higher methylation suggests poorer stress regulation or adversity exposure. This also makes it more difficult to understand the rationale for testing correlations in total methylation across all four genes.

Would it be possible for the authors to correct for genetics in their analyses, if these data are available? Knowing whether there are any confounding effects by genetics would help interpret the epigenetic findings.

The final research question on stress and methylation of the parent and child needs more detailed reporting. Please report the reliability of the composite score and the range of the scale and descriptives such as mean of SD for the sample. This will help interpretation of the results (e.g., null results could be driven by little variance and generally low scores for example but not able to see this from the paper).

Why did the author decide to perform multiple linear regression only with those genes that were significantly correlated instead of performing a multiple regression for each gene from the start? This could be clarified in a data analysis section in the methods.

The multiple regression would ideally be presented as a table either within the main paper or a supplement – seeing the total variance explained by the model, the F value, and the standard errors will help readers understand the results and the authors’ interpretation.

The finding that all but one association between stress and methylation were non-significant seems surprising and warrants more discussion. What do the authors think might explain this?

More minor comments, mainly regarding reporting:

  • In the results the heading for research question 4 seems to be missing
  • The method section is missing a data analysis section. Please report the software used for the analyses (program and packages) and a description of the steps taken in the analysis. E.g. how missing data was handled?
  • The authors mention that they corrected for multiple testing with Bonferroni correction during the discussion but do not mention this earlier in the methods or result section. The authors could specifiy how the correction was applied in the methods section and report whether each result passes correction in the results and tables. Care should be taken when summarizing the findings of the study if not all results passed correction (i.e.,. in the abstract and discussion).
  • Bonferroni corrected significance would be good to include in tables.
  • I’m missing a table with characteristics – e.g. mean and SDs for methylation values and mean stress scores. This would help the reader interpret the results and evaluate the interpretation.
  • Reliabilty of the composite child stress score made from the two measures needs reporting and arguably it would be wise to standardize the scores before summing to create the composite (e.g. converting to Z scores) – or if this was already done report it.

Two additional strengths I would like to highlight:

  • Testing child age as moderator was an interesting addition with a sound rationale and good explanation of the null results.
  • Although writing needs to be more nuanced in several places (see earlier comment), writing is very clear—allowing a broad audience to understand the authors’ rationale, results and interpretation.  

Author Response

Responses to Reviewer 1

The authors report on cross-sectional associations between methylation of four stress-regulation related genes (5HTT, 14 NR3C1, FKBP5, and BDNF) in mothers and their children (9-16 yrs). In addition to basic correlations, the authors report on differential associations by child age and parent- and child perceived stress. Findings will be useful tor researchers studying epigenetics of parenting and child stress regulation. As such, the paper offers a modest but nonetheless interesting expansion on existing literature.
We are grateful for the positive evaluation of our study and paper and for the helpful suggestions about which we agree that they helped to significantly strengthen the paper. Below, we will illustrate point by point how we addressed your comments.
1. Throughout the manuscript, statements need to be checked on accuracy, precision, and nuance. A few examples:
1a) “to remedy all of the shortcomings” sounds perhaps a little to absolute. This study too, focuses on a limited set of genes, and its sample is relatively small. Perhaps “to remedy some of the shortcomings” would be more accurate.
We fully agree that this is a more accurate phrasing, and this was changed accordingly (line 75).
1b) shared environment is not necessarily the most obvious explanation for concurrence. Biological dyads are likely to share genetics and this can drive methylation similarities given that methylation is also associated with genotype. More equal weighting could be given to the various explanations presented.
We would like to thank the reviewer for pointing this out. To address this thoughtful suggestion, we added this argument to that section of the introduction, which now reads as follows (line 58):
“In addition, variation in methylation levels differs depending on genotypes which parents and children share (e.g., Lim, Lin, & Karnani, 2017). This enhances the likelihood that their methylation levels covary.”
1c) The authors state they found “limited evidence that methylation were correlated with stress levels” In my view the evidence was less than limited, there was hardly any.
We agree that that formulation would better reflect the current results. We now wrote (line 291):
“However, there was hardly any support for a correlation between methylation levels and stress.”
2. In prior literature, higher methylation is not associated with poorer stress regulation and/or adversity exposure at all four genes (specifically FKBP5). It thus is not correct to state that in general higher methylation suggests poorer stress regulation or adversity exposure. This also makes it more difficult to understand the rationale for testing correlations in total methylation across all four genes.
In response to this comment, we went through the literature again. Literature does show that methylation of FKBP5 can be linked to stress dysregulation. For example, in 2019, Kang et al. found that the genetic and epigenetic modulation of FKBP5 is involved in the pathophysiology of PTSD. In another study, Mulder et al., (2017) found stronger cortisol stress responses to stress in resistant attached children with FKBP5 rs1360780 if they displayed higher levels of methylation. Although we agree, based on the literature that it is wise to be more careful in our descriptions. Therefore, we made the following changes in the manuscript:
Line 87:

“Methylation of the genes that manage this stress-regulation system can modulate levels of cortisol in response to stress, which translates into variation in self-regulation (Meaney & Szyf, 2005).”

Line 103:

“For all the above-mentioned genes, theory suggests that more exposure to adverse environments leads to higher levels of methylation which may modulate the regulation negative affect and self-regulation (Kular & Kular, 2018).”

Kang, J. I., Kim, T. Y., Choi, J. H., So, H. S., & Kim, S. J. (2019). Allele-specific DNA methylation level of FKBP5 is associated with post-traumatic stress disorder. Psychoneuroendocrinology, 103, 1–7. https://doi.org/10.1016/j.psyneuen.2018.12.226

Mulder, R. H., Rijlaarsdam, J., Luijk, M. P., Verhulst, F. C., Felix, J. F., Tiemeier, H., Bakermans-Kranenburg, M. J., & Van Ijzendoorn, M. H. (2017). Methylation matters: FK506 binding protein 51 (FKBP5) methylation moderates the associations of FKBP5 genotype and resistant attachment with stress regulation. Development and psychopathology, 29(2), 491–503. https://doi.org/10.1017/S095457941700013X

3. Would it be possible for the authors to correct for genetics in their analyses, if these data are available? Knowing whether there are any confounding effects by genetics would help interpret the epigenetic findings.
Unfortunately, we do not have such data. We mentioned this now as a separate limitation to the study in the limitations section (line 375):
“Also, we could not correct for genetics in the analyses. Genotype can affect the likelihood that genes are methylated. Therefore, it might be that we could not find associations between stress and methylation because methylation levels and methylation covariance were driven by genetic covariance (Lim et al., 2017), an explanation we cannot rule out in the current study because we had no genetic data”
4. The final research question on stress and methylation of the parent and child needs more detailed reporting. Please report the reliability of the composite score and the range of the scale and descriptives such as mean of SD for the sample. This will help interpretation of the results (e.g., null results could be driven by little variance and generally low scores for example but not able to see this from the paper).
When rereading the paper, we could not but agree that expanding more on the stress-related analyses would strengthen the paper. Regarding the issues mentioned in the comment, we can say that we first evaluated whether it would be relevant to sum the scores. We did bivariate correlation analyses and found that all the scores correlated significantly (***, p < .001):
1 2 3
1. ALEQ 1
2. YLSI_severity .54*** 1
3. YLSI_chronic .46*** .60*** 1

Hence it is unlikely that our null-findings reflect the impact of error in our stress-measure.
We further reported the M and SD for the sum score and for the parent ALEQ score. For the child sum score: scores ranged from 46 to 209; M = 157.79; SD = 22.48. Moreover, data distribution was skewed: Skewness = -1.13, SE(Skewness) = .19; Kurtosis = 5.10, SE(Kurtosis) = .38. Removing participants from the correlation analyses when they had scores that differed more than 2SDs from the mean, further reduced the correlations between the child stress sum score and the methylation scores and turned the association between child stress and BDNF non-significant.

For the parent stress score: scores ranged from 0 to 17; M = 4.58; SD = 3.42. Moreover, data distribution was skewed: Skewness = 86, SE(Skewness) = .19; Kurtosis = .44, SE(Kurtosis) = .38. Removing participants from the correlation analyses when they had scores that differed more than 2SDs from the mean, further reduced the correlations between the mother stress score and the methylation scores

Hence, in line with the intuition of the reviewer, it is not unlikely that restriction of range might have affected the correlations, which is also relevant to mention in the discussion of the null-findings.

To respond to this comment, we added all the requested extra information.
First, we added information on the correlation between the child stress scores to the methods section as follows (line 216):
“The three indicators of stress in the child were highly correlated (.46 < r < .60; ps < .001) suggesting that it was better to combine all measures to reduce the number of analyses.”
Second, we added information on the means and distribution of the stress scores (line 264):
“Preliminary inspection of the stress scores, showed that they had a skewed distribution. For mothers’ stress: M = 4.58; SD = 3.42; Skewness = 86, SE(Skewenss) = .19; Kurtosis = .44, SE(Kurtosis) = .38. For children’s stress: M = 157.79; SD = 22.48. Moreover, data distribution was skewed: Skewness = -1.13, SE(Skewenss) = .19; Kurtosis = 5.10, SE(Kurtosis) = .38.”
Third, we reported that removing participants diverging more than 2 SDs from the mean on the stress measure reduced the one significant correlation between BDNF and child stress to non-significance (line 271).
“This correlation was no longer significant after excluding scores differing more than 2SDs from the child stress mean, r = .05, p = .51.”
Finally, as suggested by the reviewer, we mentioned that one reason why we hardly found a correlation between stress and methylation could be lack of variation (line 312):
“One reason for the lack of correlations between the stress measures and the methylation measures might be lack of variance at the level of stress. Supporting this interpretation, the one observed association between child stress and BDNF methylation disappeared when removing more extreme scores.”
5. Why did the author decide to perform multiple linear regression only with those genes that were significantly correlated instead of performing a multiple regression for each gene from the start? This could be clarified in a data analysis section in the methods.
Our decision was based on the idea that when stress is not related to child or mother methylation levels, it is hard to theoretically assume that stress explains the mother-child methylation covariance. Because we only found for BDNF an association between stress and child methylation levels, we only reported that one analysis. However, we can just as easily report all the multiple regression analyses. In line with our theoretical argument, for all the genes, mother and child methylation levels continued to be significantly associated. This was now added to the paper (line 286):
“Second, we used multiple regression analysis to test whether mother-child methylation covariance survived controlling for mother and child stress. The mother-child methylation associations remained significant for all the analyses: 5HTT: R2 = .06, F(3, 152) = 2.98, p = .03, with 5HTTmother = .20, p = .04; NR3C1: R2 = .13, F(3, 152) = 7.62, p < .001, with NR3C1mother = .33, p < .001; FKBP5: R2 = .03, F(3, 152) = 4.10, p = .008, with FKBP5mother = .16, p = .044; BDNF: R2 = .08, F(3, 152) = 7.62, p < .001, with BDNFmother = .15, p = .43; Overall level of Methylation: R2 = .16, F(3, 152) = 9.94, p < .001, with methylation_mother = .36, p < .001. ”
6. The multiple regression would ideally be presented as a table either within the main paper or a supplement – seeing the total variance explained by the model, the F value, and the standard errors will help readers understand the results and the authors’ interpretation.
As requested, per multiple regression analysis, we added R2 and F. For now, we did not add SE as we reported standardized regression weights, so SE did not seem to add much to the result section. Off course, if requested, we are more than willing to add this information as well.
7. The finding that all but one association between stress and methylation were non-significant seems surprising and warrants more discussion. What do the authors think might explain this?
There are many possible explanations mentioned in the paper. We added, in response to this reviewer’s request 1b, that genetic covariance could account for epigenetic covariance, but that we could not test this. Therefore, we now mentioned this as a limitation to the study (line 422).
“Also, we could not correct for genetics in the analyses. Genotype can affect the likelihood that genes are methylated and therefore methylation covariance might be driven by genetic covariance (Lim et al., 2017), an explanation we cannot rule out in the current study.”
An alternative explanation might be that we measured child methylation levels established earlier in life (here we also discuss the epigenetic clock as requested by reviewer 3, comment 4). As a result, it might have been impossible for methylation levels to correlate with the current levels of stress.
Also, research on environmental exposure and methylation shows robust effects for chemical exposure (e.g., Martin & Fry, 2018). There is little doubt that also non-chemical exposure can be linked to methylation as well (most famous are the licking and grooming studies in rodents, e.g., Bagot et al., 2012), but in human samples, it proves challenging to find link between stress and methylation levels is hard (e.g., Rijlaarsdam et al., 2016).
Finally, we mentioned here genetic covariance and epigenetic inheritance as explanations we could not test in the current study:
“However, there was hardly any support for a correlation between methylation levels and stress. One reason for the lack of correlations between the stress measures and the methylation measures might be lack of variance at the level of stress in the current sample. Supporting this interpretation, the one observed association between child stress and BDNF methylation disappeared when removing more extreme scores. Moreover, ours is not the first study that struggled to find clear correlations between stress and methylation levels in humans (e.g., Rijlaarsdam et al., 2016). In contrast to what is known about the epigenetic impact of chemical exposures, more research is needed to understand how and when stress exposure affects methylation (Martin & Fry, 2018). For example, it might be that the shared methylation within each mother-child dyad we found in the current study reflects the ongoing effect of shared past exposure to chemical exposures (Murgatroyd & Spengler, 2011). Furthermore, recent research on the epigenetic clock (e.g., Suarez et al., 2018) and on the Developmental Origins on Health and Disease model (Heindel & Vandenberg, 2015) suggest that methylation levels are to a substantial extent established earlier in life, making it harder to find significant correlations between ongoing stress and methylation levels at later ages. Also, we could not correct for genetics in the analyses. Genotype can affect the likelihood that genes are methylated. Therefore, it might be that we could not find associations between stress and methylation because methylation levels and methylation covariance were driven by genetic covariance (Lim et al., 2017), an explanation we cannot rule out in the current study because we had no genetic data. Harder to test but impossible to rule out is the possibility that the covariance could also reflect epigenetic inheritance (Heard & Martienssen, 2014).”

Bagot, R.C., Zhang, T.Y., Wen, X.L., Nguyen, T.T.T., Nguyen, B.H., Diorio, J., Wong, T.P., & Meaney, M.J. (2012). Variations in postnatal maternal care and the epigenetic regulation of Grm1 expression and hippocampal function in the rat. Proceedings of the National Academy of Sciences, 109, 17200-17207.

Martin, E. M., & Fry, R. C. (2018). Environmental influences on the epigenome: exposure-associated DNA methylation in human populations. Annual review of public health, 39, 309-333.

Rijlaarsdam, J., Pappa, I., Walton, E., Bakermans-Kranenburg, M. J., Mileva-Seitz, V. R., Rippe, R. C., ... & van IJzendoorn, M. H. (2016). An epigenome-wide association meta-analysis of prenatal maternal stress in neonates: A model approach for replication. Epigenetics, 11(2), 140-149.
8. More minor comments, mainly regarding reporting:
8a) In the results the heading for research question 4 seems to be missing

We are grateful the reviewer noticed this omission, we added the heading. (pg. 6)

8b) The method section is missing a data analysis section. Please report the software used for the analyses (program and packages) and a description of the steps taken in the analysis. E.g. how missing data was handled?

We added a data analysis section in which we also reported which software was used for the analyses. (Pg. 4 line 196)
“All analyses were conducted using SPSS, version 27. First, we conducted preliminary analyses to evaluate whether child gender and age were directly linked to child methylation levels. Then, for Research Question 1, we calculated correlations to assess mother-child methylation covariance per gene. For Research Question 2, we used Principal Component Analysis (PCA) on the four gene’s methylation scores for children and mothers separately. These PCAs allowed to calculate a methylation component score per member of the dyad to assess with a bivariate correlation analysis whether there is a general association between mother and child methylation independent of the specific genes. For Research Question 3, we evaluated the moderating effect of age on the association between mother and child methylation levels using Hayes's (2013) PROCESS tool for SPSS (Model 1). Separate analyses were performed for each individual gene and for the methylation component score. For Research Question 4, we first conducted correlation analyses to evaluate the association between maternal stress and maternal methylation levels on the one hand and between child stress and child methylation levels on the other hand. Finally, we conducted multiple regression analyses to evaluate the association between maternal and child methylation levels while controlling for the mother and child stress scores. ”

We discussed missing data and how we handled missing data in the results section: (line 216)

“No methylation data was missing. For 30 children, interview data was partially incomplete. When calculating the sum score, we used mean imputation. After that, less than 3 percent of the data was missing (1 child stress score, 4 parent stress scores), which were pairwise deleted.”

8c) The authors mention that they corrected for multiple testing with Bonferroni correction during the discussion but do not mention this earlier in the methods or result section. The authors could specifiy how the correction was applied in the methods section and report whether each result passes correction in the results and tables. Care should be taken when summarizing the findings of the study if not all results passed correction (i.e.,. in the abstract and discussion).

We added the requested information. As this comment also seemed to imply that it was important to apply Bonferroni correction to the correlation analyses we conducted between mother and child methylation levels for each individual gene. This was added for Research Question 1: (pg.5 line 226)
“However, Bonferroni correction for multiple testing requires p-values to be lower than .0125 (/4 tests) due to which the transgenerational correlations for FKBP5 and BDNF no longer reached significance (p = .044 and p = .046 respectively).”

As requested by the reviewer, we also adjusted the discussion in line with this finding: (line284)

“At the level of single genes, methylation in all four genes significantly correlated across the mothers and the children. However, the correlations for FKBP5 and BDNF did not survive Bonferroni correction. The results of the current study regarding FKBP5 were weaker than what Yehuda et al., (2016) observed in Holocaust survivors. One reason why the size of the effect found in the current study was smaller, could be the fact that the current study’s dyads were not exposed to the extreme distress that Holocaust survivors experienced. Nevertheless, the trend-like effect we found in the current study is in line with Yehuda’s study showing transgenerational covariance between mother and child methylation levels. Moreover, the current study added to the literature by investigating mother-child methylation covariance for additional genes and by investigating covariance at the same point in time. In line with expectations, methylation levels correlated between generations (mother–child) for HTT5 and NR3C1.”

This required, as assumed by the reviewer, an adjustment to the abstract:

“Results showed that mother and offspring 5HTT and NR3C1 methylation patterns correlated.”

For Research Question 4, we added the following information to the results section as requested: (pg.6 line 265)

“However, Bonferroni correction for multiple testing requires p-values to be lower than .01 (/5 tests) due to which the correlation between BDNF methylation and child stress no longer reached significance (p = .025).”

8d) Bonferroni corrected significance would be good to include in tables.

For now, we opted to explain the correction in detail in the results section and not include it in the tables (see response to 8c). If deemed necessary, we can also add this information to a footnote to the table.

8e) I’m missing a table with characteristics – e.g. mean and SDs for methylation values and mean stress scores. This would help the reader interpret the results and evaluate the interpretation.

We added this information to the text. If deemed necessary, we can also add this information as a table. (line 258)

“Preliminary inspection of the stress scores, showed that they had a skewed distribution. For mothers’ stress: M = 4.58; SD = 3.42; Skewness = 86, SE(Skewness) = .19; Kurtosis = .44, SE(Kurtosis) = .38. For children’s stress: M = 157.79; SD = 22.48. Moreover, data distribution was skewed: Skewness = -1.13, SE(Skewness) = .19; Kurtosis = 5.10, SE(Kurtosis) = .38.”

8f) Reliabilty of the composite child stress score made from the two measures needs reporting and arguably it would be wise to standardize the scores before summing to create the composite (e.g. converting to Z scores) – or if this was already done report it.

As requested, we added reliability analyses: (line 239)
The adequacy of combining methylation levels across genes was further supported by reliability analysis (Cronbach  children = .72; Cronbach  mothers: .87).
For now, we did not standardize the scores for each individual gene before conducting PCA as PCA builds on standardized regression.
9) Two additional strengths I would like to highlight:
9a) Testing child age as moderator was an interesting addition with a sound rationale and good explanation of the null results.
9b) Although writing needs to be more nuanced in several places (see earlier comment), writing is very clear—allowing a broad audience to understand the authors’ rationale, results and interpretation.

We would like to thank the reviewer for highlighting these strengths.

Reviewer 2 Report

This is a well written and structured manuscript.

My only objection is  that authors did not present data on the correlation between the levels of stress hormon cortisol and the pattern of DNA methylation 

Author Response

This is a well written and structured manuscript.

We are grateful for these words of praise.

My only objection is  that authors did not present data on the correlation between the levels of stress hormon cortisol and the pattern of DNA methylation 

We agree that if we had had cortisol stress response data, this would have added weight to the findings. Therefore, we mentioned this now in the limitations section on line 332:

Third, we could not present valid information on the mothers' and children's self-regulatory capacity nor on the participants’ cortisol responses to stress. Such information would have further strengthened the point that transgenerational epigenetic covariance might explain part of the transmission stress-regulation-related difficulties in parent-child dyads.

Reviewer 3 Report

Overall this is a very good paper. It has a clear and well defined hypothesis, appropriate analytical methods and is overall a well written, well described paper. I have only minor concerns: 

  1. The wording of how the PCA analysis is described should be changed. The way it is described in the introduction section makes it seem like all the genes were tested together in a way that does have a biological reasoning behind it. The test is not really of concordance but a test of covariance: PCAs are used to described in the effect of all variables is greater than each one individually. Given the analysis is sound, rewording the intro should be sufficient to correct this issue. 
  2. A statistical methods section describing the software used, types of testing schemes with an accompanying description of the the relevance given the study design, and multiple test corrections should be added. 
  3. what exactly does it mean when the the "overall level" is described? Typically this type of language is used to describe global methylation levels, which don't seem to be assessed. Is this the principle components?  This is very unclear. 
  4. A discussion of these genes relative to the sites found in the epigenetic clock should be included as the HPA axis and GR signaling are enriched within the at least one of the clocks that has been developed. 
  5. The article "Environmental Influences on the Epigenome: Exposure- Associated DNA Methylation in Human Populations" by Martin and Fry may be helpful to these authors in terms of a description of the biology of environmental exposures and the relationship to changes in methylation. Additionally, the developmental origins of health and disease hypothesis, and the specific timeline of epigenetic programming may be of interest to the authors and could help explain some of the lack of association between discordance of epigenetic profiles and age. 

Author Response

Overall this is a very good paper. It has a clear and well defined hypothesis, appropriate analytical methods and is overall a well written, well described paper. I have only minor concerns: 

We would like to thank the reviewer for this positive evaluation of our study and paper. We are grateful for the helpful suggestions and agree that they served to strengthen the paper.

  1. The wording of how the PCA analysis is described should be changed. The way it is described in the introduction section makes it seem like all the genes were tested together in a way that does have a biological reasoning behind it. The test is not really of concordance but a test of covariance: PCAs are used to described in the effect of all variables is greater than each one individually. Given the analysis is sound, rewording the intro should be sufficient to correct this issue. 

We understand that we created confusion about the nature of our analyses to frame the study as a test of concordance. Therefore, throughout the paper, from the title until the end of the discussion, we systematically replaced the word concordance with covariance as suggested by the reviewer.

  1. A statistical methods section describing the software used, types of testing schemes with an accompanying description of the the relevance given the study design, and multiple test corrections should be added. 

We agree that such a section would contribute to the paper. So, we added a section “plan of the analysis”: (Pg. 4 line 196)

“All analyses were conducted using SPSS, version 27. First, we conducted preliminary analyses to evaluate whether child gender and age were directly linked to child methylation levels. Then, for Research Question 1, we calculated correlations to assess mother-child methylation covariance per gene. For Research Question 2, we used Principal Component Analysis (PCA) on the four gene’s methylation scores for children and mothers separately. These PCAs allowed to calculate a methylation component score per member of the dyad. Then, we assessed with a bivariate correlation analysis whether there is a general association between mother and child methylation independent of the specific genes. For Research Question 3, we evaluated the moderating effect of age on the association between mother and child methylation levels using Hayes's (2013) PROCESS tool for SPSS (Model 1). Separate analyses were performed for each individual gene and for the methylation component score. For Research Question 4, we first conducted correlation analyses to evaluate the association between maternal stress and maternal methylation levels on the one hand and between child stress and child methylation levels on the other hand. Finally, we conducted multiple regression analyses to evaluate the association between maternal and child methylation levels while controlling for the mother and child stress scores. ”

Also, we added multiple test corrections in each relevant analysis. Concretely, for research question 1: (pg.5 line 226)

 However, Bonferroni correction for multiple testing requires p-values to be lower than .0125 (/4 tests) due to which the transgenerational correlations for FKBP5 and BDNF no longer reached significance (p = .044 and p = .046 respectively).

And for research question 4: (pg.6 line 265)

“However, Bonferroni correction for multiple testing requires p-values to be lower than .01 (/5 tests) due to which the correlation between BDNFmethylation and child stress no longer reached significance (p = .025). Moreover, this correlation was no longer significant after excluding scores differing more than 2SDs from the child stress mean, r = .05, p = .51.”

  1. what exactly does it mean when the the "overall level" is described? Typically this type of language is used to describe global methylation levels, which don't seem to be assessed. Is this the principle components?  This is very unclear. 

We understand that we created confusion using the words “overall level”, so we decided to stay in our language closer to the analysis on which we based these scores. In line with this reviewer’s suggestion, we talk now about “methylation component” and “methylation component score” throughout the paper.

  1. A discussion of these genes relative to the sites found in the epigenetic clock should be included as the HPA axis and GR signaling are enriched within the at least one of the clocks that has been developed. 

We are grateful to the reviewer for pointing us to this interesting literature. We mentioned it in the discussion (see also response to comment 5).

  1. The article “Environmental Influences on the Epigenome: Exposure- Associated DNA Methylation in Human Populations” by Martin and Fry may be helpful to these authors in terms of a description of the biology of environmental exposures and the relationship to changes in methylation. Additionally, the developmental origins of health and disease hypothesis, and the specific timeline of epigenetic programming may be of interest to the authors and could help explain some of the lack of association between discordance of epigenetic profiles and age. 

Again, we are grateful for these useful suggestions. In response to this comment and comment 4, we now added this to the discussion: (pg.5 line 316).

“In contrast to what is known about the epigenetic impact of chemical exposures, more research is needed to understand how and when stress exposure affects methylation (Martin & Fry, 2018). For example, it might be that the shared methylation within each mother-child dyad we found in the current study reflects the ongoing effect of shared past exposure to chemical exposures (Murgatroyd & Spengler, 2011). Furthermore, recent research on the epigenetic clock (e.g., Suarez et al., 2018) and on the Developmental Origins on Health and Disease model (Heindel & Vandenberg, 2015) suggest that methylation levels are to a substantial extent established earlier in life, making it harder to find significant correlations between ongoing stress and methylation levels at later ages. Also, we could not correct for genetics in the analyses.”